# C-to-N atom swapping and skeletal editing in indoles and benzofurans

Zhe Wang[1], Pengwei Xu[1,2], Shu-Min Guo[1,2], Constantin G. Daniliuc[1] & Armido Studer[1✉]

Skeletal editing comprises the structural reorganization of compounds. Such editing can be achieved through atom swapping, atom insertion, atom deletion or reorganization of the compound's backbone structure[1,2]. Conducted at a late stage in drug development campaigns, skeletal editing enables diversification of an existing pharmacophore, enhancing the efficiency of drug development. Instead of constructing a heteroarene classically from basic building blocks, structural variants are readily accessible directly starting from a lead compound or approved pharmacophore. Here we present C to N atom swapping in indoles at the C2 position to give indazoles through oxidative cleavage of the indole heteroarene core and subsequent ring closure. Reactions proceed through ring-opened oximes as intermediates. These ring deconstructed intermediates can also be diverted into benzimidazoles resulting in an overall C to N atom swapping with concomitant skeletal reorganization. The same structural diverting strategies are equally well applicable to benzofurans leading to either benzisoxazoles or benzoxazoles. The compound classes obtained through these methods—indazoles[3,4], benzisoxazoles[5], benzimidazoles[6,7] and benzoxazoles[8]—are biologically relevant moieties found as substructures in natural products and pharmaceuticals. The procedures introduced substantially enlarge the methods portfolio in the emerging field of skeletal editing.

Indoles are among the most important heteroarenes in chemistry, characterized by a planar bicyclic structure consisting of a benzene and a fused pyrrole ring (Fig. 1a). The indole moiety, a highly potent pharmacophore, serves as the core structure in many drugs and is also present in various natural products[9–11]. Substituting the NH group in indoles with an oxygen atom yields benzofurans, another significant class of heteroarenes. As oxygen analogues of indoles, benzofurans show valuable biological activities and are commonly found as substructures in natural products and pharmaceuticals[12,13].

Skeletal editing of heteroarenes has recently emerged as a highly valuable tool for altering the core structure of these molecules at a late stage[1,2]. Instead of constructing a heteroarene from basic building blocks, skeletal editing enables diversification of an existing complex pharmacophore, thereby enhancing the efficiency of drug development[14–20]. Essentially, the synthesis begins with the final product. Skeletal editing can be achieved through atom swapping, atom insertion, atom deletion or reorganization of the compound's backbone structure. For example, the Vasil'ev and Park groups disclosed oxygen-to-nitrogen transmutations of isoxazoles to pyrazoles and furans to pyrroles, respectively[21,22]. Herein we report C to N atom swaps in indoles and benzofurans to give indazoles[23,24] and benzisoxazoles[5]. Both these bicyclic heteroarenes show interesting and highly valuable biological activities. Moreover, by using a slightly varied chemical strategy, benzimidazoles[25] and benzoxazoles[8] are equally well accessible starting from the same intermediates. These latter two important compound classes are formed through C to N atom swapping with concomitant skeletal reorganization. During the preparation of this manuscript, Morandi and coworkers reported a similar C to N swap approach of indoles to benzimidazoles through sequential oxidation with subsequent Beckmann rearrangement[26]. This strategy operates orthogonal to ours while selectively transforming 2,3-unsubstituted indoles. However, indole-to-indazole conversion as well as the skeletal editing of benzofurans was not reported by the authors.

Skeletal editing of indoles has recently garnered significant interests. The research groups of Levin[27], Ball[28], Xu[29] and Glorius[30] have achieved elegant carbon insertion, leading to the formation of quinolines (Fig. 1b). N insertion into indoles, as demonstrated by Morandi[31,32] and Ackermann[33], provides direct access to quinazolines. In the context of skeletal editing of benzofurans, Yorimitsu successfully accomplished the insertion of boron, silicon, germanium, phosphorous and titanium into the C2–O bond[34,35]. However, skeletal editing of indoles and benzofurans to give all four classes of title compounds is currently unknown and certainly highly desirable.

Our general design strategy for the structural reorganization of both indoles and benzofurans is presented in Fig. 1c. On oxidative cleavage of the C2=C3 double bond in these heteroarenes, oximes should be accessible. Transformation of the OH group of the oxime functionality into a leaving group will afford reactive intermediates that are poised to engage in a Beckmann rearrangement to give nitrilium cations. Intramolecular trapping of such cations by the N respective O atom of the former indole or benzofuran core should directly afford the corresponding benzimidazoles and benzoxazoles, respectively[36,37]. On the other hand, the activated oxime ether functionality might also

[1]Organisch-Chemisches Institut, Universität Münster, Münster, Germany. [2]These authors contributed equally: Pengwei Xu, Shu-Min Guo. ✉e-mail: studer@uni-muenster.de

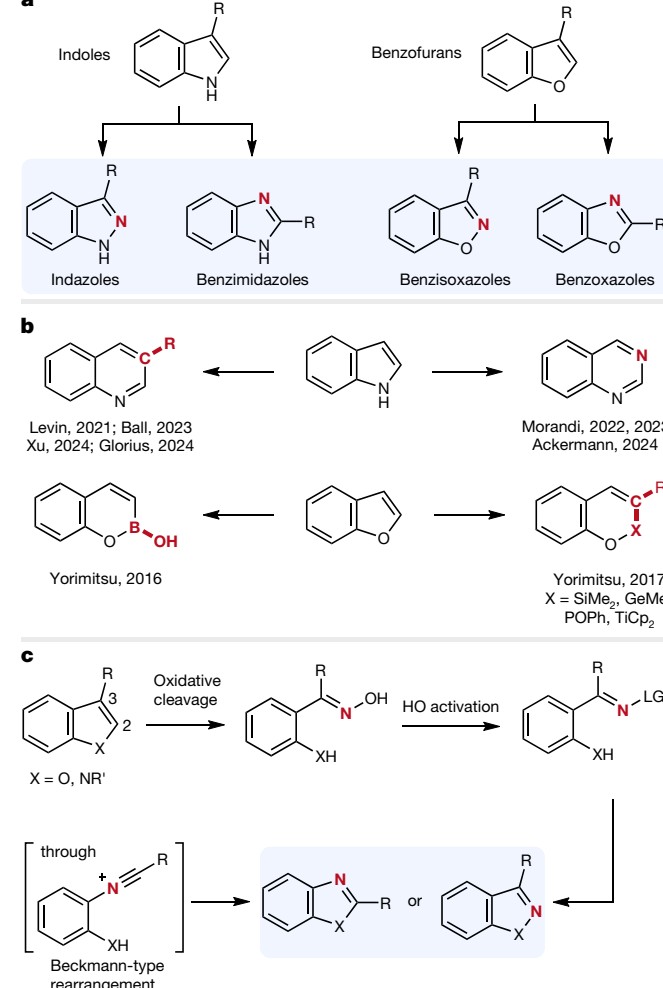

**a**

Indoles

Benzofurans

Indazoles

Benzimidazoles

Benzisoxazoles

Benzoxazoles

**b**

Levin, 2021; Ball, 2023
Xu, 2024; Glorius, 2024

Morandi, 2022, 2023
Ackermann, 2024

Yorimitsu, 2016

Yorimitsu, 2017
X = SiMe$_2$, GeMe$_2$
POPh, TiCp$_2$

**c**

Oxidative cleavage

HO activation

X = O, NR'

through

Beckmann-type
rearrangement

**Fig. 1 | Skeletal editing of indoles and benzofurans. a**, Transformation of indoles to indazoles and benzimidazoles requires C to N atom swapping and also skeletal reorganization for the latter process. Analogous transformations on benzofurans deliver benzisoxazoles and benzoxazoles. **b**, State of the art in skeletal editing of indoles and benzofurans. These known processes all proceed through atom insertion resulting in ring enlargement. **c**, Design of a reaction sequence for the editing of indoles and benzofurans without changing the ring size. These structural reorganizations comprise oxidative cleavage of the heteroarene entity, OH-activation followed by Beckmann-type rearrangement or substitution at nitrogen. LG, leaving group.

act as an N electrophile to directly provide indazoles or benzisoxazoles through N or O addition onto the oxime nitrogen atom[38,39].

Studies were commenced by investigating the ring cleavable radical oximation of the N-protected indole Moc-**1a** to give **I-1**. N-nitrosamines are known to engage in light-mediated N−N bond homolysis to give the persistent NO radical along with an aminyl radical. Under acidic conditions, the latter gets protonated to a more reactive aminyl radical cation, which efficiently adds to alkenes and the adduct transient C radicals can then be trapped by the persistent NO to give the corresponding β-amino alkyl nitroso compounds[40]. We tested this transformation on indole Moc-**1a** using N-nitrosomorpholine as the aminyl radical precursor. Reaction is best conducted in ethyl acetate by irradiation with a blue light-emitting diode (415 nm, 3 W) in the presence of toluene sulfonic acid for 48 h at room temperature to give **I-1** in high yield (78%) (Fig. 2a). We propose that the aminyl radical cation generated through light-mediated N−N bond homolysis and protonation reacts at the C2 position of Moc-**1a** to give the distonic benzylic radical cation **A** that is selectively trapped by the persistent NO (ref. 41) to give the nitroso

adduct **B** (Fig. 2b). This nitroso compound **B** is suggested to engage in a C−C bond cleavage with concomitant proton transfer to give oxime **C** (ref. 42). Hydrolysis eventually leads to the intermediate **I-1**.

With the optimized protocol for mild radical cleavage of the indole core in hand, we next focused on the diversification towards construction of indazoles and benzimidazoles. Under Mitsunobu conditions[38] the N-protected indazole Moc-**2a** was obtained in 64% overall yield. On the other hand, O-mesylation of the oxime **I-1** with methanesulfonyl chloride followed by heating gave the benzimidazole Moc-**3a** through a Beckmann-type rearrangement in 61% overall yield[36]. We noted that N protection of the indole is required for successful oxidative radical cleavage of the indole core. Therefore, an alternative protocol was developed that is also working on free NH indoles. Oxidative cleavage of the C2=C3 double bond by means of Witkop−Winterfeldt oxidation[43,44] and subsequent deformylation led to the corresponding ortho-acyl aniline **I-2**, which on treatment with Me$_3$C$_6$H$_2$SO$_2$ONHBoc directly afforded the free indazole **2a** in good overall yield (Fig. 2c)[39]. Transformation of ortho-acyl aniline to a benzimidazole can be realized on N-protected anilines. Thus, Witkop−Winterfeldt oxidation on indole **1a**, N protection with 4-toluenesulfonyl chloride, o-aminoaryl ketimine formation with ammonia and PhI(OAc)$_2$ mediated Beckmann-type rearrangement provided the N-protected benzimidazole Ts-**3a** (58%)[37]. We were pleased to find that oxidative radical cleavage with N-nitrosamines is also applicable to benzofurans, thereby significantly expanding our strategy, as documented by the successful skeletal editing of benzofuran **4a** to give benzisoxazole **5a** through C to N atom swapping (73%) (Fig. 2d). As for the indole series, the oxime intermediate **I-3** also acts as a common editing platform for the benzofurans and accordingly **I-3** can be readily diverted to also access the corresponding benzoxazole **6a** (63%). Alternatively, a complementary ionic oxidative cleavage of the benzofuran ring is also provided. The C2=C3 double bond of benzofuran **4m** is cleaved on treatment with pyridinium chlorochromate (PCC) and subsequent aminolysis gives the respective o-hydroxyaryl ketimine **I-4**. Chlorination of the imine with N-chlorosuccinimide under basic conditions or with aqueous NaOCl smoothly forms benzisoxazole **5m** (55%) or the corresponding benzoxazole **6m** (83%), respectively[45].

The scope of these different transformations was investigated on various indoles and benzofurans using either the radical-mediated approach (method A) or an ionic oxidative cleavage of the heteroarene core with different oxidants (method B). In selected cases both methods were tested (Figs. 3 and 4). Electron-donating substituents such as methoxy, methyl or benzyloxy at the 5- or 6-position of the indole entity in 3-methyl substituted systems were tolerated and as examples the indazoles Moc-**2b** and Moc-**2d** were obtained in 46−60% yield using the radical approach. The corresponding benzimidazoles Moc-**3b** and Moc-**3d** as well as Moc-**3f** were equally well accessible. Steric effects are of importance, as the 4-methyl indole **1c** gave the edited indazole **2c** and benzimidazole Ts-**3c** in lower yields. We found that an electronic withdrawing group (cyano or halo-substituents) at the indole core significantly reduced reactivity for the radical cleavage, and the targeted indazoles were obtained in low yields along with unreacted starting indoles (**2e** and Boc-**2g**). The size of the 3-alkyl substituent on the indole also affects reactivity. Whereas indoles bearing secondary or tertiary carbon substituents in the 3-position (cyclohexyl **1i**, 2-adamantyl **1j** or cumyl **1k**) are smoothly converted to indazoles **2i**−**2k** in 38−64% yield, the corresponding imidazoles were formed with lower efficiency (Ts-**3i** and Ts-**3k**). Compared to those congeners, the sec-butyl indole reacted less efficiently through the radical pathway. Yet, the targeted indazole Moc-**2h** and benzimidazole Moc-**3h** could be readily isolated and unreacted starting Moc-indole **1h** was recovered in both cases. We were pleased to find that 3-aryl-substituted indoles also engaged in these skeletal editing processes, as documented for the phenyl derivative **1l**. The indazole **2l** was obtained in 68% yield, whereas Ts-**3l** was isolated in 30%. In the latter case, indazole or benzimidazole selectivity for the final heteroarene ring reconstruction

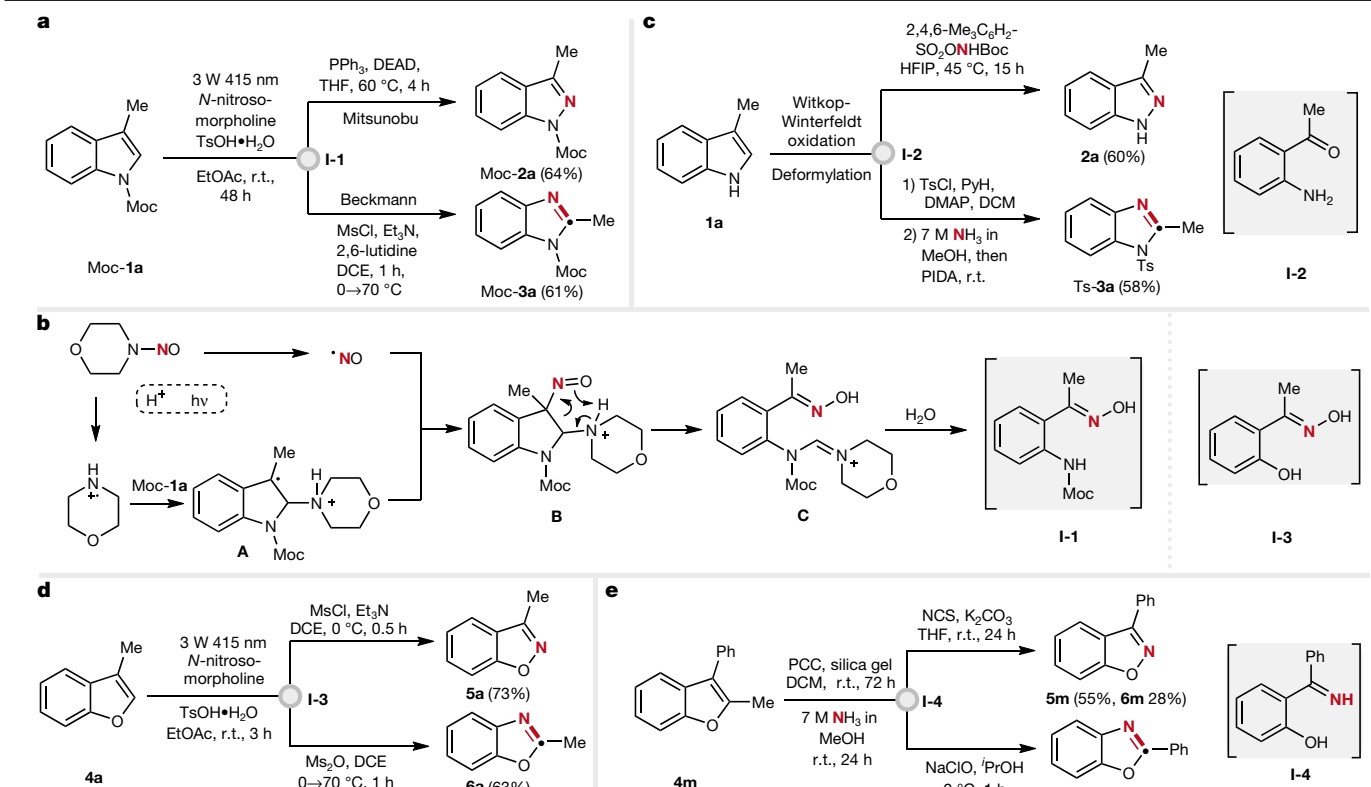

**Fig. 2 | Experimental realization of the diverting structural editing of indoles and benzofurans to give indazoles, benzimidazoles, benzisoxazoles and benzoxazoles. a**, Radical 1,2-aminonitrosylation of a 3-substituted indole to give a ring-opened oxime that can further react to an indazole or a benzimidazole, respectively. **b**, Proposed mechanism for the radical oxidative ring-opening of the heteroarene entity in an indole. **c**, Alternatively, oxidative ring cleavage of the indole heteroarene core can also be achieved through Witkop–Winterfeldt oxidation and subsequent deformylation to give the corresponding *o*-aminoaryl ketone that is further converted to an

indazole or to a ketimine giving a benzimidazole. **d**, Analogous chemical transformations on benzofurans. **e**, Alternative protocol for transforming benzofurans to benzisoxazoles and benzoxazoles. Moc, methoxycarbonyl; Ts, tosyl; HFIP, hexafluoroisopropanol; MsCl, methanesulfonyl chloride; Ms₂O, methanesulfonic anhydride; PIDA, (diacetoxyiodo)benzene; NCS, *N*-chlorosuccinimide; r.t., room temperature; PCC, pyridinium chlorochromate; DCM, dichloromethane; DCE, dichloroethane; DMAP, 4-dimethylaminopyridine; DEAD, diethyl azodicarboxylate; THF, tetrahydrofuran.

was not perfect and we obtained the N-tosylated indazole Ts-**2l** as the byproduct (33%), showing that the selective reorganization is not that trivial. The parent unsubstituted indole was successfully converted to indazole **2m** (41%). For the 2,3-disubstituted indoles **1n** and **1o** we successfully achieved the CMe and CCF₃ to N swap showing that this structural editing is not restricted to 2-unsubstituted indoles. The indazole **2a** and the benzimidazole Ts-**3a** were both formed in good yields from parent **1n**. However, the trifluoromethyl congener **1o** afforded the respective products **2a** and **Ts-3a** in lower yield (24–26%) due to the electron-withdrawing effect of the CF₃ group lowering reactivity for the initial oxidative cleavage[46]. To further showcase the use of our strategy, challenging 2,3-disubstituted indoles bearing six- or seven-membered fused rings (**1q** and **1p**) were also examined inspired by the prevalence of polycyclic indole scaffolds in nature. Six-membered ring annellated derivative **1q** could be transformed through Witkop–Winterfeldt oxidation and subsequent Schmidt-type rearrangement to benzimidazole **3q'** bearing an enlarged seven-membered ring in good yield (54%). However, the seven-membered ring congener **1p** reacted with lower efficiency to **3p'** (24%), probably due to inherent eight-membered ring strain that was built up during structural reorganization. Notably, **1q** can also be transformed to indazole **2q** and benzimidazole Ts-**3q** carrying ring-opened linear chain substituents.

We next addressed the editing of benzofurans and noted generally higher efficiency in the radical-mediated oxidative cleavage process compared to the indoles (Fig. 4). The reaction of the 5-acetoxy-substituted 3-methylbenzofuran (**4b**) provided benzisoxazole **5b** in high yield (74%)

and the corresponding reorganized benzoxazole **6b** was isolated in 80% yield. The 5-propargyloxy-substituted benzofuran reacted equally well to give **5c** and **6c** in good yields. It is important to note that the triple bond in **4c** is tolerated under the radical conditions. Further, prenyloxy-benzofurans are also eligible substrates as shown by the successful preparation of **5d**, albeit a lower yield was achieved. A trifluorosulfonyloxy substituent is compatible with our two processes and the targeted **5e** and **6e** were obtained in good yields. We could also show that the benzannellated benzofuran **4f** was readily converted to its edited benzisoxazole **5f** and benzoxazole **6f**. Benzofurans featuring different substitution patterns on the 3-position as well as 2,3-disubstitued benzofurans were also evaluated. The trifluoroacetamide substituted benzofuran **4g** could be transformed into the corresponding benzoxazole **6g** in excellent yield (82%). However, the benzisoxazole **5g** was obtained in diminished yield (23%) caused by unproductive N–N interactions between the side chain amide and the oxime moiety during the ring closure step under the basic conditions. Silyl ether and phosphine oxide were compatible under the reaction conditions affording benzisoxazole **5h** and benzoxazole **6j** in good yields (62–68%).

When a cyanomethyl group was installed at the 3-position, the oxime group intermediately formed in the radical-mediated oxidative cleavage of benzofuran **4i** subsequently attacked as a nucleophile the cyano moiety to eventually give the isoxazole **5i** (58%). To compare the relative efficiency of the radical-induced oxidative cleavage between indoles and benzofurans, substrate **4k** bearing both of these heteroarene cores was transformed, showcasing larger reactivity for the

**Fig. 3 | Scope of the structural editing of indoles.** All reactions were conducted under the optimized conditions specified in Fig. 2. For further details we refer to the Supplementary Information. Yields provided correspond to isolated overall yields starting from the corresponding indole. Method A corresponds to the radical-induced heteroarene ring cleavage with *N*-nitrosomorpholine, and method B uses different chemical oxidants to cleave the indole ring. [a]Yield of the recovered starting material.

benzofuran moiety, which was efficiently edited to the benzoxazole **6k** (50%) with the indole core remaining unreacted. CMe to N swap in 2,3-dimethylbenzofuran **4l** was successfully realized to give **5a** and **6a** through the radical pathway. Moreover, the sterically more demanding disubstituted benzofurans **4m** and **4n** also engaged in CMe or CPh to N swap using PCC for initial oxidative cleavage. Benzofuran **4o** carrying a six-membered fused ring was successfully converted to the open-chain benzisoxazole **5o** (46%) and benzoxazole **6o** (40%).

Finally, we tested our methods on a wide range of more complex compounds (Fig. 5). The naproxen-indole conjugate Moc-**1r** was successfully transformed to the indazole Moc-**2r** (15%) and its isomeric benzimidazole Moc-**3r** (15%) under the radical conditions, whereas unreacted starting material could be recovered in a roughly 50% yield. Similar outcomes were obtained for indole Moc-**1s** and Moc-**1t** that were prepared from indole-3-propionic acid and febuxostat. Using the alternative oxidant-mediated protocol (method B) significantly improved yields were obtained for the overall sequences. Thus, the indoles **1r**–**1t** were successfully edited to provide the indazoles **2r**–**2t** in 46–67% yield and the corresponding benzimidazoles Ts-**3r** and Ts-**3s** were obtained in 47–54% yield. Furthermore, the developed strategies were also applicable to the late-stage diversification of bioactive compounds and drug-derived structures. Protected tryptophan **1u** gave the corresponding indazole product **2u** in good yield (58%). Likewise, tryptophan-containing dipeptide **1v** and tripeptide **1w** were converted smoothly to indazoles **2v** and **2w**. Brevianamide F (**1x**), which shows diverse biological activities and acts as important starting material for different fumitremorgin class alkaloids[47], was structurally reorganized to afford the corresponding indazole **2x** and benzimidazole Ts-**3x**. Tryptamines are found in the human brain as important neurotransmitters and represent core structures of many drugs[48], so we tested the skeletal editing on derivative **1y** to afford indazoles Eoc-**2y** and **2y**, as well as the benzimidazoles Eoc-**3y** and Ts-**3y** under both radical and the chemical oxidant-induced ring-cleavage conditions (Eoc, ethoxycarbonyl). Better yields for this substrate were obtained through the ionic pathway (method A Eoc-**2y** (24%) versus method B **2y** (60%); method A Eoc-**3y** (25%) versus method B Ts-**3y** (34%)). Similarly, indole **1z**, which is a potent 5-lipoxygenase inhibitor[49], was successfully converted to indazole **2z** (56%) as well as benzimidazole Ts-**3z** (36%), further expanding the functional group tolerance towards amides. Noteworthy, the antibacterial agent[50] **1aa** bearing both a tethered pyridine core and a sterically demanding isopropyl substituent at the C3-position was successfully transformed to the indazole **2aa** documenting the tolerance towards pyridines. The structure of **2aa** was unambiguously assigned by X-ray crystal structure analysis. Pimprinine, an alkaloid originally isolated from *Streptomyces* showing a variety of biological activities[51], afforded the corresponding atom swapped indazole product **2ab** (38%), leaving the oxazole heterocycle untouched. Moreover, the etodolac-derived substrate **1ac** was transformed to the desired seven-membered ring-fused benzimidazole product **3ac** (46%), underlining the potential of the protocol for late-stage modification of drugs.

Along with bioactive indoles we also tested the skeletal editing of bioactive and drug-conjugated benzofurans using the radical approach.

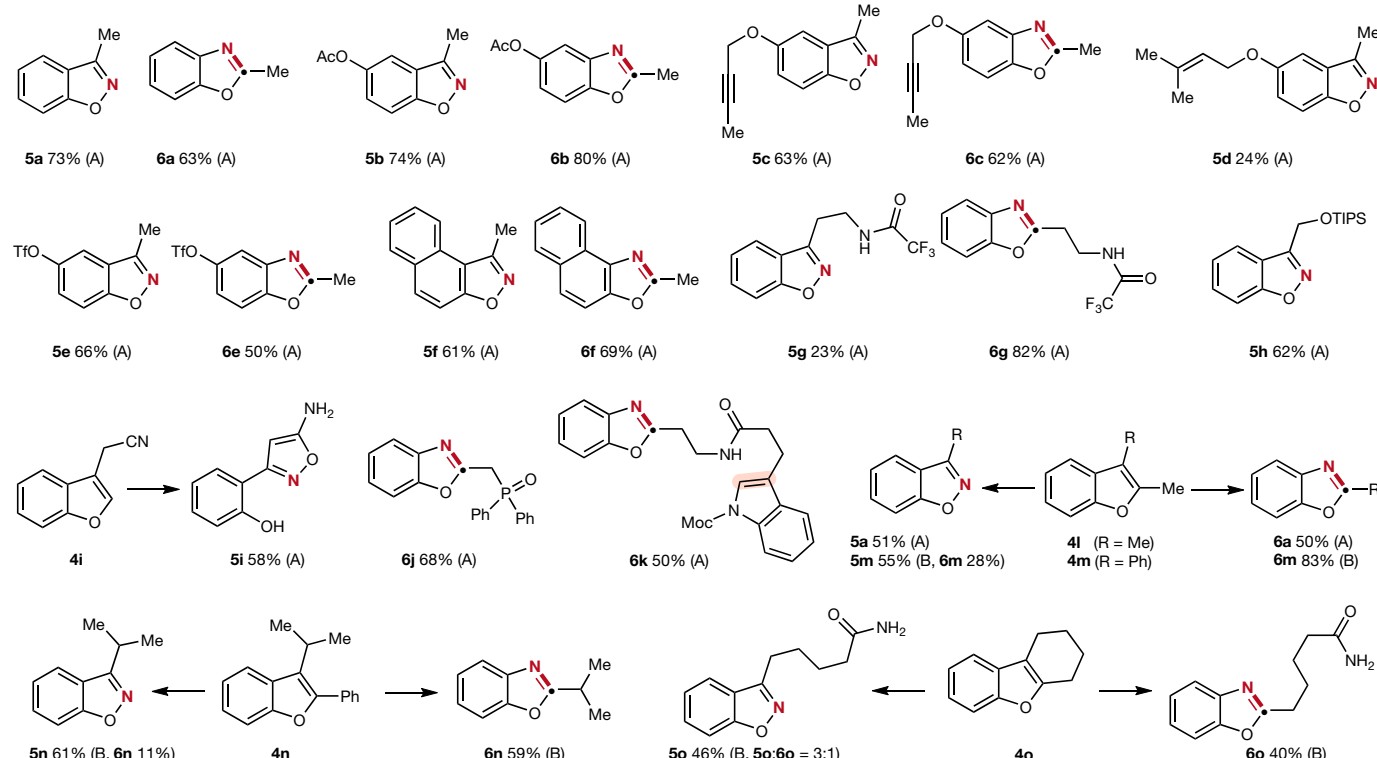

**Fig. 4 | Scope of the structural editing of benzofurans.** All reactions were conducted under the optimized conditions specified in Fig. 2. For further details we refer to the Supplementary Information. Yields provided correspond to isolated overall yields starting from the corresponding benzofuran. Method A corresponds to the radical-induced heteroarene ring cleavage with *N*-nitrosomorpholine, and method B uses PCC to oxidatively cleave the benzofuran ring. TIPS, triisopropylsilyl.

The benzofuran **4p** tethering a thioether-bridged pyridine core at the C3 positon, showing antibacterial activities in plants[52], was successfully edited to benzoxazole **6p** in 60% yield. Furthermore, the skeleton of substrate **4q**, a new derivative of the photochemotherapeutic agents methoxsalen and trioxsalen[53], gave the corresponding benzisoxazole **5q** (48%) and benzoxazole **6q** (76%) using a one-pot process that will be further described below. The benzofuran **4r** derived from oxaprozin was readily transformed to the benzisoxazole **5r** in excellent 81% yield. Applying the diverting reaction path, benzoxazole **6r** was obtained in 60% yield. Notably, the oxazole moiety in **4r** remained unreacted. We also tested a complex system derived from indomethacin bearing both an indole and a benzofuran moiety (**4s**). As expected, the sterically less hindered benzofuran entity reacted chemo-selectively to give benzisoxazole **5s** in 80% yield. Starting with the same substrate **4s**, benzoxazole **6s** was obtained in a good yield (65%). The structures of **5r** and **6s** were unambiguously assigned by X-ray crystal structure analysis.

The above-discussed diversified skeletal editing of indoles and benzofurans is highly efficient to quickly access up to four different classes of products through common intermediates. However, when no diversification and only one product is demanded efficient one-pot manipulations are often desired. Therefore, we showcased that our strategy can be also efficiently performed in one-pot. This was demonstrated with benzofuran **4s** as the starting material using an in situ formed *N*-nitrosomorpholine solution (Extended Data Fig. 1a). The corresponding benzisoxazole **5s** was obtained in 72% and the benzoxazole **6s** in 55% overall yield, the latter on a 3.0-mmol scale. Noteworthy, as *N*-nitrosomorpholine shows toxicity, its in situ formation and direct use is risk lowering. Finally, we could show that skeletal editing with concomitant [15]N-labelling is possible through the herein introduced approach. Selectively [15]N-labelled *N*-nitrosomorpholine was readily prepared from commercially available Na[15]NO$_2$ (Extended Data Fig. 1b and

Supplementary Information). With this labelled nitroso amine in hand, Moc-**1a** was successfully converted to Moc-[15]N-**2a** that was isolated in 60% yield with more than 95% [15]N-incorporation. CH to [15]N swap was also achieved on the indole Moc-**1ad** to afford the indazole Moc-[15]N-**2ad** in 26% yield along with 39% unreacted indole. The lower yield is due to the presence of the electron-withdrawing ester group at the indole ring. Of note, Moc-[15]N-**2ad** is a building block to access labelled orally bioavailable small-molecule inhibitors of CDK8 for the treatment of cancer or for the synthesis of sodium channel inhibitors[54,55].

## Online content

1. Xu, P. & Studer, A. Skeletal editing through cycloaddition and subsequent cycloreversion reactions. *Acc. Chem. Res.* **58**, 647–658 (2025).
2. Cheng, Q. et al. Skeletal editing of pyridines through atom-pair swap from CN to CC. *Nat. Chem.* **16**, 741–748 (2024).
3. Qin, J., Cheng, W., Duan, Y.-T., Yang, H. & Yao, Y. Indazole as a privileged scaffold: the derivatives and their therapeutic applications. *Anticancer Agents Med. Chem.* **21**, 839–860 (2021).
4. Pennington, L. D. & Moustakas, D. T. The necessary nitrogen atom: a versatile high-impact design element for multiparameter optimization. *J. Med. Chem.* **60**, 3552–3579 (2017).
5. Rakesh, K. P., Shantharam, C. S., Sridhara, M. B., Manukumar, H. M. & Qin, H.-L. Benzisoxazole: a privileged scaffold for medicinal chemistry. *Medchemcomm* **8**, 2023–2039 (2017).
6. Tyagi, Y. K., Jali, G. & Singh, R. Synthesis and anti-cancer applications of benzimidazole derivatives—recent studies. *Anticancer Agents Med. Chem.* **22**, 3280–3290 (2022).
7. Monga, J. et al. Unlocking the pharmacological potential of benzimidazole derivatives: a pathway to drug development. *Curr. Top. Med. Chem.* **24**, 437–485 (2024).
8. Wong, X. K. & Yeong, K. Y. A patent review on the current developments of benzoxazoles in drug discovery. *ChemMedChem* **16**, 3237–3262 (2021).

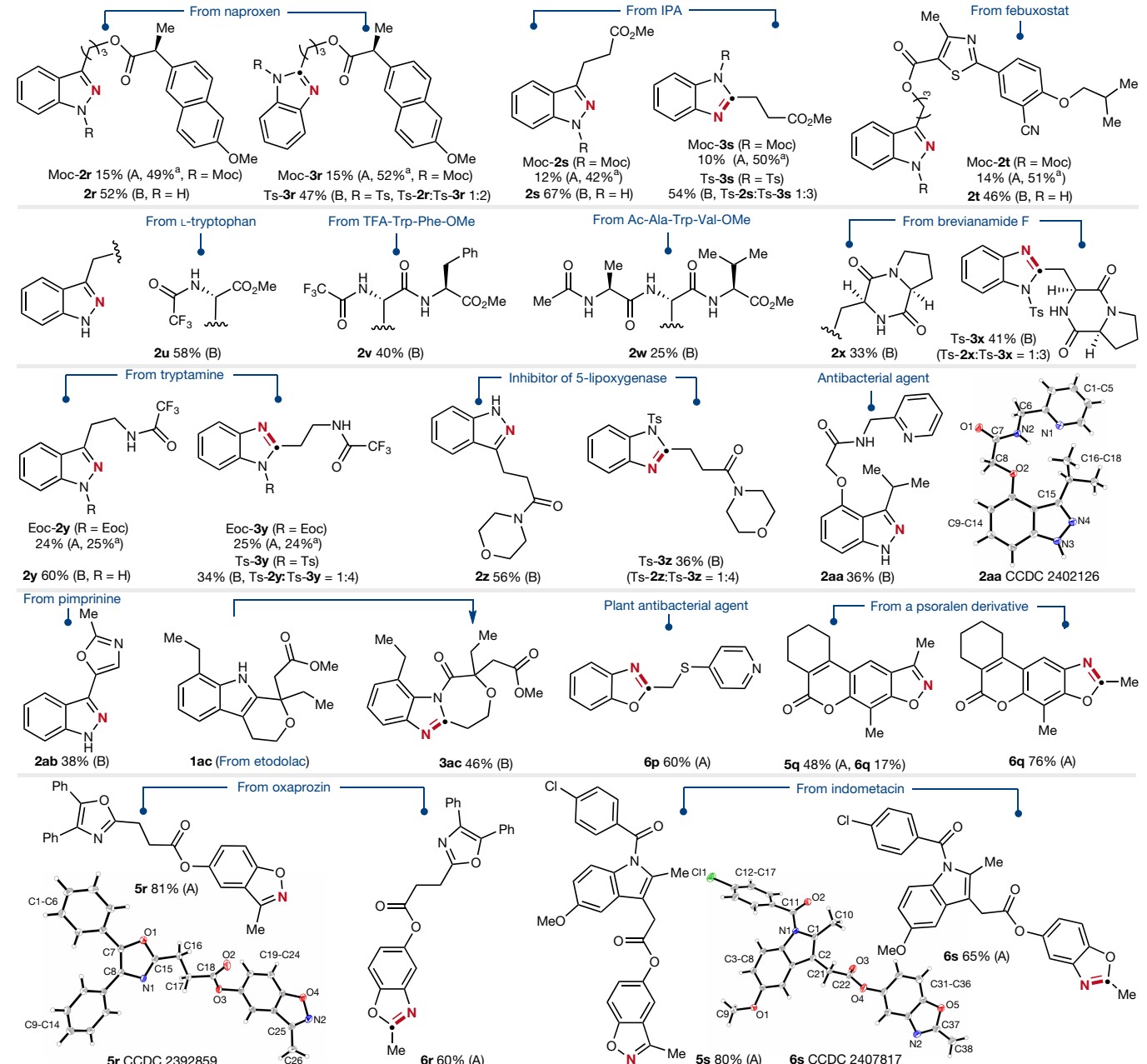

**Fig. 5 | Structural editing of drugs or drug-like compounds that contain an indole or benzofuran core structures.** Skeletal editing of indoles and benzofurans with tethered bioactive moieties and pharmaceutical compounds. Method A corresponds to the radical-induced heteroarene ring cleavage with

*N*-nitrosomorpholine, and method B uses ionic oxidants to cleave the heterocycle rings. For further details, refer to the Supplementary Information. ªYield of the recovered starting material. IPA, indole-3-propionic acid.

9. Sravanthi, T. V. & Manju, S. L. Indoles—a promising scaffold for drug development. *Eur. J. Pharm. Sci.* **91**, 1–10 (2016).
10. Sarkara, N. et al. Insights of indole: a novel target in medicinal chemistry (a review). *Russ. J. Gen. Chem.* **93**, 1791–1841 (2023).
11. Singh, T. P. & Singh, O. M. Recent progress in biological activities of indole and indole alkaloids. *Mini Rev. Med. Chem.* **18**, 9–25 (2018).
12. Dwarakanath, D. & Gaonkar, S. L. Advances in synthetic strategies and medicinal importance of benzofurans: a review. *Asian J. Org. Chem.* **11**, e202200282 (2022).
13. Miao, Y.-H. et al. Natural source, bioactivity and synthesis of benzofuran derivatives. *RSC Adv.* **9**, 27510–27540 (2019).
14. Fan, Z. et al. Molecular editing of aza-arene C–H bonds by distance, geometry and chirality. *Nature* **610**, 87–93 (2022).
15. Jurczyk, J. et al. Single-atom logic for heterocycle editing. *Nat. Synth.* **1**, 352–364 (2022).
16. Joynson, B. W. & Ball, L. T. Skeletal editing: interconversion of arenes and heteroarenes. *Helv. Chim. Acta* **106**, e2022001 (2023).
17. Liu, Z., Sivaguru, P., Ning, Y., Wu, Y. & Bi, X. Skeletal editing of (hetero)arenes using carbenes. *Chem. Eur. J.* **29**, e202301227 (2023).
18. Peplow, M. Skeleton crew. *Nature* **618**, 21–24 (2023).
19. Ma, C., Lindsley, C. W., Chang, J. & Yu, B. Rational molecular editing: a new paradigm in drug discovery. *J. Med. Chem.* **67**, 11459–11466 (2024).
20. Xu, Y.-A., Xiang, S.-H., Che, J.-T., Wang, J.-B. & Tan, B. Skeletal editing of cyclic molecules using nitrenes. *Chin. J. Chem.* **42**, 2656–2667 (2024).
21. Sviridov, S. I., Vasil'ev, A. A. & Shorshnev, S. V. Straightforward transformation of isoxazoles into pyrazoles: renewed and improved. *Tetrahedron* **63**, 12195–12201 (2007).
22. Kim, D. et al. Photocatalytic furan-to-pyrrole conversion. *Science* **386**, 99–105 (2024).
23. Buchi, G., Lee, G. C. M., Yang, D. & Tannenbaum, S. R. Direct acting, highly mutagenic, α-hydroxy N-nitrosamines from 4-chloroindoles. *J. Am. Chem. Soc.* **108**, 4115–4119 (1986).
24. Chevalier, A., Ouahrouch, A., Arnaud, A., Gallavardin, T. & Franck, X. An optimized procedure for direct access to 1H-indazole-3-carboxaldehyde derivatives by nitrosation of indoles. *RSC Adv.* **8**, 13121–13128 (2018).
25. Leonori, D. et al. Photochemical conversion of indazoles into benzimidazoles. *Angew. Chem. Int. Ed.* https://doi.org/10.1002/anie.202423804 (2025).
26. Paschke, A.-S. et al. Carbon-to-nitrogen atom swap enables direct access to benzimidazoles from drug-like indoles. Preprint at https://doi.org/10.26434/chemrxiv-2024-prwm8 (2024).

27. Dherange, B. D., Kelly, P. Q., Liles, J. P., Sigman, M. S. & Levin, M. D. Carbon atom insertion into pyrroles and indoles promoted by chlorodiazirines. *J. Am. Chem. Soc.* **143**, 11337–11344 (2021).

28. Joynson, B. W., Cumming, G. R. & Ball, L. T. Photochemically mediated ring expansion of indoles and pyrroles with chlorodiazirines: synthetic methodology and thermal hazard assessment. *Angew. Chem. Int. Ed.* **62**, e202305081 (2023).

29. Guo, H., Qiu, S. & Xu, P. One-carbon ring expansion of indoles and pyrroles: a straightforward access to 3-fluorinated quinolines and pyridines. *Angew. Chem. Int. Ed.* **63**, e202317104 (2024).

30. Wu, F.-P., Tyler, J. L., Daniliuc, C. G. & Glorius, F. Atomic carbon equivalent: design and application to diversity-generating skeletal editing from indoles to 3-functionalized quinolines. *ACS Catal.* **14**, 13343–13351 (2024).

31. Reisenbauer, J. C., Green, O., Franchino, A., Finkelstein, P. & Morandi, B. Late-stage diversification of indole skeletons through nitrogen atom insertion. *Science* **377**, 1104–1109 (2022).

32. Reisenbauer, J. C. et al. Direct access to quinazolines and pyrimidines from unprotected indoles and pyrroles through nitrogen atom insertion. *Org. Lett.* **25**, 8419–8423 (2023).

33. Zhang, B.-S. et al. Electrochemical skeletal indole editing via nitrogen atom insertion by sustainable oxygen reduction reaction. *Angew. Chem. Int. Ed.* **63**, e202407384 (2024).

34. Saito, H., Otsuka, S., Nogi, K. & Yorimitsu, H. Nickel-catalyzed boron insertion into the C2–O bond of benzofurans. *J. Am. Chem. Soc.* **138**, 15315–15318 (2016).

35. Tsuchiya, S., Saito, H., Nogi, K. & Yorimitsu, H. Manganese-catalyzed ring opening of benzofurans and its application to insertion of heteroatoms into the C2–O bond. *Org. Lett.* **19**, 5557–5560 (2017).

36. Wray, B. C. & Stambuli, J. P. Synthesis of N-arylindazoles and benzimidazoles from a common intermediate. *Org. Lett.* **12**, 4576–4579 (2010).

37. Zhang, X., Huang, R., Marrot, J., Coeffard, V. & Xiong, Y. Hypervalent iodine-mediated synthesis of benzoxazoles and benzimidazoles via an oxidative rearrangement. *Tetrahedron* **71**, 700–708 (2015).

38. Conlon, I. L., Konsein, K., Morel, Y., Chan, A. & Fletcher, S. Construction of 1*H*-indazoles from ortho-aminobenzoximes by the Mitsunobu reaction. *Tetrahedron Lett.* **60**, 150929 (2019).

39. Wang, J. et al. A scalable and metal-free synthesis of indazoles from 2-aminophenones and in situ generated de-boc-protected *O*-mesitylsulfonyl hydroxylamine derivatives. *J. Org. Chem.* **88**, 13049–13056 (2023).

40. Chow, Y. L. Photo-addition of N-nitrosodialkylamines to cyclohexene. *Can. J. Chem.* **43**, 2711–2716 (1965).

41. Leifert, D. & Studer, A. The persistent radical effect in organic synthesis. *Angew. Chem. Int. Ed.* **59**, 74–108 (2020).

42. Chow, Y. L., Colon, C. & Chen, S. C. Photochemistry of nitroso compounds in solutions. VII. Photoaddition of nitrosamines to various olefins. *J. Org. Chem.* **32**, 2109–2115 (1967).

43. Breinbauer, R. & Mentel, M. The Witkop–Winterfeldt-oxidation of indoles. *Curr. Org. Chem.* **11**, 159–176 (2007).

44. Dolby, L. J. & Booth, D. L. The periodate oxidation of indoles[1]. *J. Am. Chem. Soc.* **88**, 1049–1051 (1966).

45. Chen, C., Andreani, T. & Li, H. A divergent and selective synthesis of isomeric benzoxazoles from a single N–Cl imine. *Org. Lett.* **13**, 6300–6303 (2011).

46. Xu, J., Liang, L., Zheng, H., Chi, Y. R. & Tong, R. Green oxidation of indoles using halide catalysis. *Nat. Commun.* **10**, 4754 (2019).

47. Maiya, S., Grundmann, A., Li, S. & Turner, G. The fumitremorgin gene cluster of *Aspergillus fumigatus*: identification of a gene encoding brevianamide F synthetase. *ChemBioChem* **7**, 1062–1069 (2006).

48. Singh, Y. P. & Kumar, H. Tryptamine: a privileged scaffold for the management of Alzheimer's disease. *Drug Dev. Res.* **84**, 1578–1594 (2023).

49. Zheng, M. et al. Indole derivatives as potent inhibitors of 5-lipoxygenase: design, synthesis, biological evaluation, and molecular modeling. *Bioorg. Med. Chem. Lett.* **17**, 2414–2420 (2007).

50. Comas, M. D. C. S. et al. Indole derivatives and their use as antibiotics. European patent EP2639220A1 (2013).

51. Liu, B. et al. Discovery of pimprinine alkaloids as novel agents against a plant virus. *J. Agric. Food Chem.* **67**, 1795–1806 (2019).

52. Jiang, S. et al. Facile access to benzofuran derivatives through radical reactions with heteroatom-centered super-electron-donors. *Nat. Commun.* **14**, 7381 (2023).

53. Garazd, Ya. L., Garazd, M. M., Shilin, S. V. & Khilya, V. P. Modified coumarins. 3. Psoralen and allopsoralen analogs. *Chem. Nat. Compd.* **37**, 409–420 (2001).

54. Li, Y. et al. Discovery of potent, selective, and orally bioavailable small-molecule inhibitors of CDK8 for the treatment of cancer. *J. Med. Chem.* **66**, 5439–5452 (2023).

55. Bell, A. S. et al. Indazole derivatives as sodium channel inhibitors. UK patent WO2012095781A1 (2012).

## Data availability

Details on the experimental procedures and all analytical data of the compounds prepared are available in the Supplementary Information. Crystallographic data for the structures reported in this Article have been deposited at the Cambridge Crystallographic Data Centre, under deposition numbers CCDC 2402126 (**2aa**), 2392859 (**5r**) and 2407817 (**6s**). Copies of the data can be obtained free of charge at https://www.ccdc.cam.ac.uk/structures/.

**Acknowledgements** We thank the China Scholarship Council (fellowship to Z.W.), Alexander von Humboldt foundation (fellowship to S.-M.G.) and the Deutsche Forschungsgemeinschaft for supporting this work. We thank J. Lammert (University of Münster) for the synthesis of some substrates and M. Haring (University of Münster) for repeating one reaction.

**Author contributions** Z.W. and A.S. conceptualized the work. Z.W. and A.S. conceived and designed the experiments. Z.W., S.-M.G. and P.X. performed the experiments and analysed the data. C.G.D. performed the X-ray analysis. A.S. wrote the manuscript.

**Funding** Open access funding provided by Universität Münster.

**Competing interests** The authors declare no competing interests.

### Additional information
**Correspondence and requests for materials** should be addressed to Armido Studer.

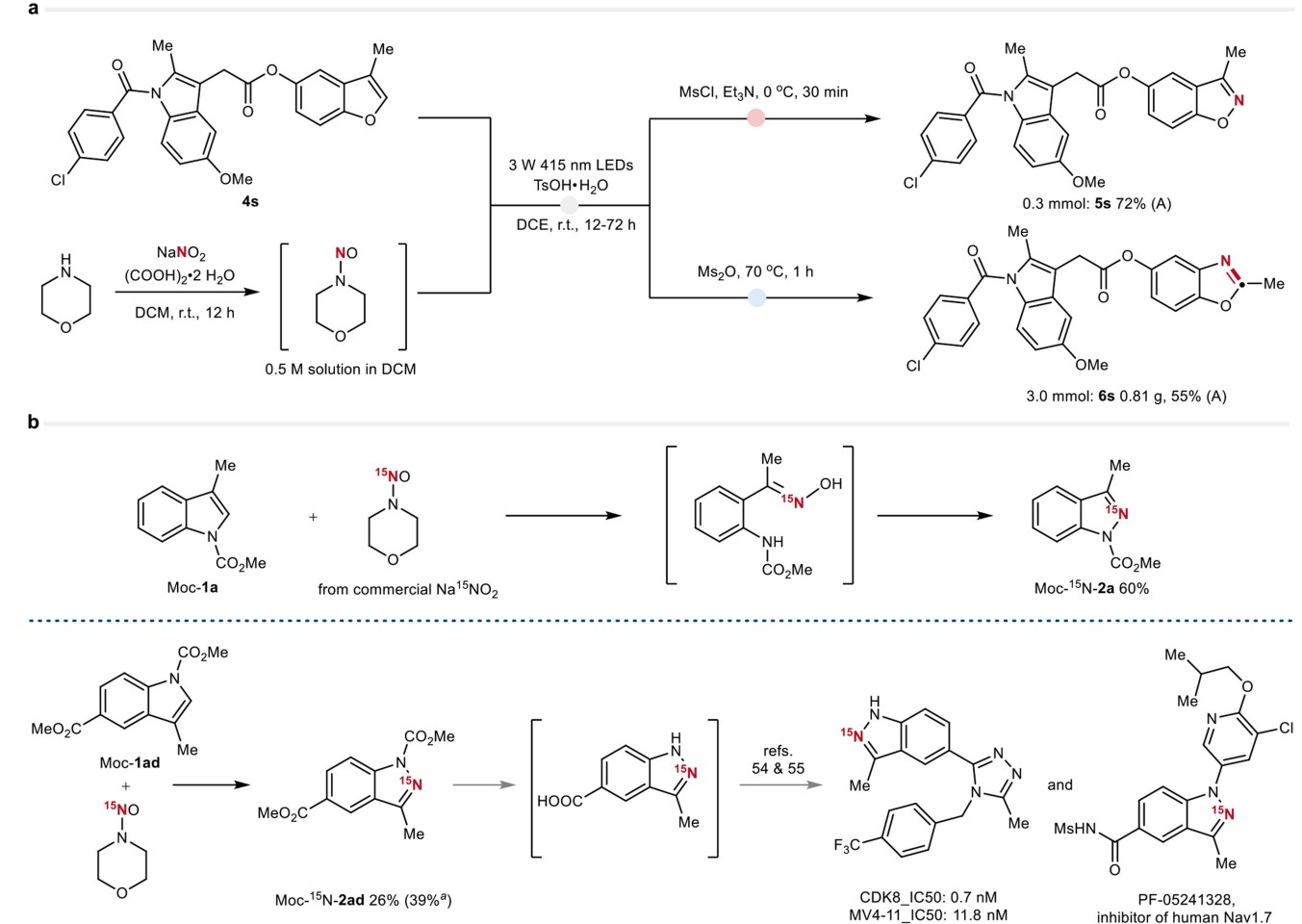

**Extended Data Fig. 1 | Synthetic applications. a**. Skeletal editing of benzofurans as one-pot processes starting from morpholine and larger scale synthesis. **b**. Synthesis of [15]N-labeled bioactive compounds through late stage C to [15]N atom swap.