## [Peer Review File · Nature]

C-to-N atom swapping and skeletal editing in indoles and benzofurans

Corresponding Author: Professor Armido Studer

Version 0:

Reviewer comments:

Referee #1

(Remarks to the Author)

In this manuscript, Studer and coworkers present an innovative approach for converting indoles and benzofurans into indazoles, benzimidazoles, benzoxazoles, and benzisoxazoles. The authors propose a stepwise strategy that involves photomediated ring-opening of benzofused five-membered heterocycles, followed by nitrogen incorporation to form new heterocyclic frameworks. This methodology seamlessly integrates photochemical reactions with subsequent nitrogen insertion, showcasing a creative synthetic design.

The applicability of the methodology is impressive, as demonstrated by the breadth of substrates investigated, including commercial pharmaceuticals and drug-like molecules. A diverse array of novel heterocyclic analogs was successfully synthesized. Notably, the transformations displayed in Figure 4 stand out as particularly impactful, highlighting the broad applicability of the method to pharmaceutically relevant compounds. While some reactions yielded moderate to low conversions (below 30% in certain cases), the high yields based on recovered starting material (brsm) are commendable and remain attractive to the medicinal chemistry community. Furthermore, the authors successfully demonstrated functional group tolerance across a range of complex molecules, further enhancing the utility of the methodology.

This work significantly broadens the scope of skeletal editing, contributing valuable insights to an emerging field. It is poised to inspire further advancements in synthetic methodology and drug discovery.

Scientific Comments:

a. While several pioneering examples were cited, more recent contributions to the field could have been discussed. For instance, skeletal editing transformations of five-membered aromatic heterocycles, such as furans-to-pyrroles and isoxazoles-to-pyrazoles, are highly relevant, particularly due to their shared mechanistic features involving oxidative ring-opening. More importantly, the recent work by Morandi and coworkers (ChemRxiv, Oct. 14, 2024) should be acknowledged. Their approach employs PIDA-mediated Witkop-type oxidation followed by a Beckmann rearrangement to achieve similar indole-to-benzimidazole conversions. The authors are encouraged to compare the strengths and limitations of their method with this contemporaneous contribution, addressing aspects such as reaction conditions, substrate scope, and functional group compatibility.

b. The substrate scope for benzofuran derivatives could be further expanded. Given the higher intrinsic reactivity of benzofurans compared to indoles, it would be valuable to include examples featuring diverse substitution patterns, particularly at the C2 position.

c. The discussion of functional group tolerance for amines and protected amino groups could be elaborated. Compounds 2p and 3p are noteworthy examples, but a broader exploration of these functionalities would enhance the relevance of the work, especially for medicinal chemists working with oxidative conditions.

d. The intermediacy of proposed species, such as 2-aminoacetophenone or intermediate I-2, could be substantiated through experimental observations. Direct evidence supporting these intermediates would bolster the mechanistic insights and strengthen the overall contribution.

e. Some minor errors in the Supporting Information require correction. For instance, the ¹³C NMR assignment for compound 2p appears incorrect, as the carbonyl carbon is reported at 274.7 ppm (q, J = 36.3 Hz), likely due to an incorrect reference standard.

f. Reporting the ¹⁵N NMR spectrum for Moc-15N-2s would add valuable data, enhancing the structural characterization of this key compound.

Referee #2

(Remarks to the Author)

Dear author(s), I have carefully examined the article titled "Skeletal Editing of Indoles and Benzofurans to Give Indazoles, Benzimidazoles, Benzoxazoles, and Benzisoxazoles." While the methodology presented does fill a gap in the existing toolbox for skeletal editing, there is a disconnect between the realities of drug discovery and the primary argument promoting this methodology for its potential application for late-stage editing during the drug development process.

Condition A involves the use of a nitrosamine reagent, which pharmaceutical companies are likely to avoid at all costs due to the heightened scrutiny from the FDA concerning nitrosamines and nitrating agents. Companies would favor a few extra steps synthesis over using such reagents, as even trace amounts pose a risk of contaminating the active pharmaceutical ingredient (API) and/or excipients.

Condition B is interesting by its simplicity; however, its practical applications are restricted due to the involvement of oxygen gas, which necessitates additional safety measures which necessitates additional safety measures as some companies require that such reactions be conducted in specialized equipment or designated laboratories which would further limit its widespread implementation.

Figure 3 uses simple substrates to illustrate the scope of the reaction, which gives desired outcome with yields from low to moderate. However the scope is limited by steric effects. Most compounds of interest for pharmaceutical companies will have more than one substituent on the benzyl group, which further restricts the applicability toward drug discovery.

Figure 4 does offer interesting applications however focuses on simpler bioactive compounds. I would contend that the key advantage of a methodology discussed in this paper present lie with its use with more complex substrates, where restarting the sequence would be time-consuming.

It is evident considerable efforts have been invested into developing this methodology and I believe that this work may be more suitably published in the Journal of the American Chemical Society (JACS). I would recommend that the authors consider positioning this paper more as an expansion of the toolbox for skeletal editing rather than as a direct methodology for drug discovery.

Referee #3

(Remarks to the Author)

Studer and coworkers report a C to N atom swapping in indoles at C2 to arrive at indazoles and at C3 to give benzimidazoles. Furthermore, benzofurans can also be converted to benzisoxazoles or benzimidazoles. N-insertion into indoles has been demonstrated by Morandi (ref. 23, 24) and Ackermann (ref. 25) Yorimitsu has inserted boron, silicon, germanium, phosphorous and titanium into the C2–O bond of benzofurans. The authors claim that less work has been done on atom swapping in indoles and benzofurans.

The design rests on oxidative cleavage of the C2–C3 bond of indoles and benzofurans and formation of a nitrilium ion or substitution on an oxime that is formed.

The oxidative chemistry hinges on the work of Y. L. Chow (Ref. 32; note the volume number indicated in the manuscript is incorrect) and Ref. 34. Alternatively, the procedure of Witkopf-Winterfeldt oxidation starts the sequence to build ortho acyl aniline. At that point, the chemistry is relatively standard (Figure 2c) for example, following studies by Stambuli etc.

While the idea/concept of atom swapping is a good one, this is a multistep protocol involving workup and solvent switches. The goal of these skeletal editing reactions should be a single step or single pot procedure that will be used by others. As it stands, this is a multistep synthesis and sets the growing field of molecular editing back because it will confuse people if this is meant to represent the forefront of the field. I am not supportive of publication in the premiere journal for science such as Nature.

Also, the work of Büchi and Tannenbaum (JACS 1986, 108, 4115) with follow up by Gallavardin and Franck (RSC Adv. 2018, 8, 13121) report very similar transformations on indoles that should, at the very least, be cited here. They also demonstrate that the concept of indole to indazole is one that has been well explored. It is used extensively already in the pharmaceutical industry even though the product has a formyl group (which is actually a useful handle).

The yields e.g., to form 2m (15%) and 3m (15%) leave a lot to be desired even though a large amount of starting materials (50%) could be recovered and the Witkopf–Winterfeldt conditions worked a little better

The ability to label with ^{15}N is a nice addition to the literature but is conceptually not new in terms of using ^{15}N labeled material which is incorporated.

Overall, the idea of this manuscript is a good one. Primarily because this is a multistep process, it falls short of the most impactful studies in this area of science and should be published in a more specialized journal. The supplementary material is in good shape.

Version 1:

Reviewer comments:

Referee #1

(Remarks to the Author)

The authors have significantly revised the manuscript and further demonstrated the broad applicability of their method. It is particularly impressive that the method performs effectively across a diverse range of substrates, including those with valuable yet vulnerable functional groups. My previous concerns and technical suggestions have been thoroughly addressed.

Regarding the original concern raised by Reviewer 2 about the use of nitrosamines as reagents, the authors have provided additional data that effectively clarifies this issue. While I recognize that some concerns regarding safety and scalability remain, it is important to note, as the authors state in their response, that this study is focused on developing new methods to aid drug discovery rather than on revolutionizing large-scale manufacturing. To further strengthen the manuscript, I encourage the authors to include an HPLC trace of the isolated product in the Supplementary Materials after a single MPLC purification, to verify that no trace amounts of nitrosamine persist in the isolated mixture.

Reviewer 3 expressed concerns regarding the multi-step process. Although the process is indeed multi-step, I believe that medicinal chemists will appreciate the protocol for two main reasons: first, the transformation reported was previously considered unattainable, and second, the overall protocol is relatively straightforward. Moreover, while the individual steps are established in the literature, the innovative combination of these steps represents a significant advancement in expanding the previously inaccessible chemical space.

Overall, the manuscript has been considerably strengthened by the authors' revisions, and the work represents a valuable contribution to the field. I commend the authors for their thoughtful responses to the reviewers' comments.

Referee #3

(Remarks to the Author)

This is a revised manuscript from Wang, Studer and coworkers. Overall, it describes a multi-step conversion of substituted indoles and benzofurans by inserting nitrogen. The sequence is oxidative cleavage of the C2,C3 bond and then substitution onto a nitrogen or Beckmann rearrangement and then engaging a nitrilium species. As stated previously, the use of nitromorpholine in the presence of acid is a nice contribution that builds on what was known previously (Ref. 40). Overall, the authors have addressed the points raised by reviewers. However, this reviewer maintains that reporting transformations that are overall multistep by involving solvent switches is something that does not advance the field. In a journal like Nature, one expects a broad readership even among synthetic chemists. The highlighted transformation, if achieved in a single operation or through sequential additions would be used by medicinal chemists (who seem to be the target of the work). However, it is very involved as described. The recommendation is to publish in a more specialised journal. Before that, the following should be considered.

- a) The title should perhaps specify that this is a multistep reaction. In my opinion, the goal of skeletal editing is to accomplish single step or single pot reactions without solvent exchanges
- b) The term molecular editing is used sometimes. This nebulous term does not make sense to this reviewer
- c) C3 unsubstituted indoles and benzofurans do not work. This should be made explicitly clear. Of course, this is a complement to the Morandi work that was recently reported in the archive, but it is important to make the limitation clear. Because there are many pharmaceutical compounds etc that possess a C3 substituent, the method can still be applied to a broad range of compounds.
- d) Figure 6: skeletal is misspelled

Referee #4

(Remarks to the Author)

The reviewers' comments and the responses to the reviewers' comments have been assessed. The authors have duly complied and added the requested references for prior and contemporaneous related work. The authors have made SI additions as requested by Reviewer 1. The authors have also substantially expanded their substrate scope tables, fulfilling the requests of Reviewer 1 and Reviewer 2. In particular, the scope of the drug-like compounds in Figure 5 makes it quite

believable that this chemistry can be applied to complex compounds, with yields that, as the authors argue, are certainly good enough for the medicinal chemistry space, where often only a 1-2 mg is need for a potency assay.

The authors have thoroughly addressed the concerns of Reviewer 2 about NO_x gases and O₂. I agree with the authors that NO_x and potential nitrosamines are of no concern in the discovery space, which is the audience toward which this paper is directed. In the process space, potential nitrosamine-forming species are avoided as much as possible, or else are extremely carefully controlled - however, this paper is not meant for the process chemistry space. Regarding O₂, however, I agree with the Reviewer and disagree with the authors - med chemists are likely not going to bother setting up reactions that require O₂ gas. However, the authors do now provide alternative oxidation protocols that avoid O₂, so I consider this concern to be adequately addressed.

I understand and agree with Reviewer 2 and 3's suggestion to a journal such as JACS. There is a lack of novelty to the chemistry reactions themselves - it is more like a very good application of a sequence of chemistry. There aren't any chemistry reactions that can be called out as "the first," broadening the reader's curiosity with respect to chemical reactivity. I actually disagree with Reviewer 3's critique that the state-of-the-art in skeletal editing should be single step/pot - I have seen single-step skeletal editing approaches that are so low-yielding, and likely are in a soup of other byproduct isomers, that I doubt anyone would use them, compared to an existing, less sexy, multi-step approach that has good yields/isolations for each step. However, for a new single- or multi-step editing approach to be published in Nature, I would look for a greater degree of chemistry novelty, on top of applicability. I feel that the novelty falls short.

Responses to the reviewer's requests:

Reviewer: 1

Comments:

In this manuscript, Studer and coworkers present an innovative approach for converting indoles and benzofurans into indazoles, benzimidazoles, benzoxazoles, and benzisoxazoles. The authors propose a stepwise strategy that involves photomediated ring-opening of benzofused five-membered heterocycles, followed by nitrogen incorporation to form new heterocyclic frameworks. This methodology seamlessly integrates photochemical reactions with subsequent nitrogen insertion, showcasing a creative synthetic design.

The applicability of the methodology is impressive, as demonstrated by the breadth of substrates investigated, including commercial pharmaceuticals and drug-like molecules. A diverse array of novel heterocyclic analogs was successfully synthesized. Notably, the transformations displayed in Figure 4 stand out as particularly impactful, highlighting the broad applicability of the method to pharmaceutically relevant compounds. While some reactions yielded moderate to low conversions (below 30% in certain cases), the high yields based on recovered starting material (brsm) are commendable and remain attractive to the medicinal chemistry community. Furthermore, the authors successfully demonstrated functional group tolerance across a range of complex molecules, further enhancing the utility of the methodology.

This work significantly broadens the scope of skeletal editing, contributing valuable insights to an emerging field. It is poised to inspire further advancements in synthetic methodology and drug discovery.

Reply: We thank reviewer #1 for the positive comments on our work.

Scientific Comments:

a. While several pioneering examples were cited, more recent contributions to the field could have been discussed. For instance, skeletal editing transformations of five-membered aromatic heterocycles, such as furans-to-pyrroles and isoxazoles-to-pyrazoles, are highly relevant, particularly due to their shared mechanistic features involving oxidative ring-opening. More importantly, the recent work by Morandi and coworkers (ChemRxiv, Oct. 14, 2024) should be acknowledged. Their approach employs PIDA-mediated Witkop-type oxidation followed by a Beckmann rearrangement to achieve similar indole-to-benzimidazole conversions. The authors are encouraged to compare the strengths and limitations of their method with this contemporaneous contribution, addressing aspects such as reaction conditions, substrate scope, and functional group compatibility.

Reply: We thank reviewer #1 for alluding to these important papers. We have included the two recent papers on skeletal editing of isoxazoles-to-pyrazoles (see new Ref. 15) and furans-to-pyrroles (see new Ref. 16) in the revised version. Further, we discussed these significant contributions in the main text: "For example, the Vasil'ev and Park groups reported each oxygen-to-nitrogen transmutation of isoxazoles to pyrazoles and furans to pyrroles, respectively.^{15,16}"

As requested, we also acknowledged Morandi's work that can be found as a ChemRxiv article on the transformation of indoles to benzimidazoles with a related strategic mindset. Their strategy is able to directly convert N-alkyl indoles to the corresponding benzimidazoles using commercial PIDA and ammonium carbamate, a reactant combination the group applied already in other skeletal editings (e.g. indole to quinazoline/quinoxaline) (Ref. 31. & Ref. 32). Regarding the scope, drug-like structures were

tested in the editing and functional group tolerance was demonstrated in the elegant Morandi work. However, the presented scope covers only the conversion of N-protected indoles. The authors circumvent this by showing a consecutive atom swap debenzoylation cascade to access NH-benzimidazole in one case. Additionally, the presented substrate scope seems to be limited to C2,C3-unsubstituted indoles and accordingly exploration of functional group tolerance was performed by examining substituents on the phenyl ring of the indole. This somehow stands in contrast to most indole drugs and natural products bearing various substituents on the C2 and C3 positions as plenty show for example structural relation to tryptophan. Instead, our protocol realized not just the skeletal editing of indoles to benzimidazoles-similar to the Morandi approach-but also the transformation of indoles to indazoles as well as the editing of benzofurans to benzisoxazoles or benzoxazoles, respectively. While Morandi and co-workers focused on drugs and drug-like molecules with substituents on the carbon cycle, we showcase the transformation of a plethora of bioactive compounds with substituents on the heterocycle part of these bicyclic heteroarenes. Thus, we consider these two methods as complementary for the editing of indoles to benzimidazoles. The following sentence was added: "During the preparation of this manuscript, Morandi and co-workers reported a similar C to N swap approach of indoles to benzimidazoles via sequential oxidation with subsequent Beckmann rearrangement.²⁶ This strategy operates orthogonal to ours while selectively transforming 2,3-unsubstituted indoles. However, indole-to-indazole conversion as well as the skeletal editing of benzofurans was not reported by the authors."

b. The substrate scope for benzofuran derivatives could be further expanded. Given the higher intrinsic reactivity of benzofurans compared to indoles, it would be valuable to include examples featuring diverse substitution patterns, particularly at the C2 position.

Reply: We appreciate the suggestion of reviewer #1 to explore more sophisticated benzofurans. During the revision we have significantly expanded the scope of benzofuran derivatives with diverse substitution patterns on the C2- and C3-position (see Figure A below or Figures 4 and 5 in the revised manuscript). Amides (4g), silyl ether (4h), cyanomethyl (4i), phosphine oxide (4j), phenyl (4m) and isopropyl (4n) moieties on the C3 position and methyl (4l and 4m) as well as phenyl (4n) residues on the C2 position are well tolerated under the developed conditions. Regarding the structural editing of C2-substituted congeners, we showed that the swap of whole fragments (CMe-to-N or CPh-to-N) is feasible. The higher intrinsic reactivity of benzofurans compared to indoles was also confirmed by the selective conversion of the benzofuran core in product 6k with the indole moiety remaining unreacted. Furthermore, benzofuran 4o bearing a fused six-membered annellated ring was straightforwardly converted to either the respective benzisoxazole 5o (46%) or benzoxazole 6o (40%) with ring-opened linear chain substituents at the C3-position. Additionally, two bioactive benzofuran derivatives 4p and 4q were also edited in good yields.

Figure A Benzofurans featuring diverse substitution patterns.

c. The discussion of functional group tolerance for amines and protected amino groups could be elaborated. Compounds 2p and 3p are noteworthy examples, but a broader exploration of these functionalities would enhance the relevance of the work, especially for medicinal chemists working with oxidative conditions.

Reply: Due to the abundance of *N*-containing functional groups in various biologically relevant structures we agree with reviewer #1's suggestion. Therefore, we examined various other examples bearing protected amino groups under our developed conditions. These included benzofurans 4g, 4k, the protected tryptophan 1u as well as dipeptide 1v, tripeptide 1w and Brevianamide F, which were all converted smoothly under our oxidative conditions to the desired products. Notable, with a cyanomethyl group at the C3-position, the oxime formed through the radical-mediated oxidative cleavage subsequently adds as an *O*-nucleophile to the nitrile functionality to give the isoxazole core (4i) bearing a free amine group in good yield (see Figure B below).

Figure B Other examples bearing protected amino groups.

d. The intermediacy of proposed species, such as 2-aminoacetophenone or intermediate I-2, could be

substantiated through experimental observations. Direct evidence supporting these intermediates would bolster the mechanistic insights and strengthen the overall contribution.

Reply: We have confirmed the formation of these key intermediates using NMR spectroscopy, like 2-aminoacetophenone I-2, oxime intermediate I-3 and ketimine I-4 (see the revised SI, pages S34 and S35). The analytical data are in agreements with those reported in the literature. Therefore, we are confident regarding the proposed mechanisms.

e. Some minor errors in the Supporting Information require correction. For instance, the ^{13}C NMR assignment for compound 2p appears incorrect, as the carbonyl carbon is reported at 274.7 ppm (q, $J = 36.3$ Hz), likely due to an incorrect reference standard.

Reply: We thank reviewer #1 for carefully examining our Supporting Information. We have corrected the wrong ^{13}C NMR assignment for compound 2p and additionally double-checked the whole document and corrected additional errors found.

f. Reporting the ^{15}N NMR spectrum for Moc-15N-2s would add valuable data, enhancing the structural characterization of this key compound.

Reply: Thanks, for full structural characterization we included the ^{15}N NMR spectrum of Moc-15N-2ad (former Moc-15N-2s) in the SI on page S79.

Reviewer: 2

Comments:

Dear author(s), I have carefully examined the article titled "Skeletal Editing of Indoles and Benzofurans to Give Indazoles, Benzimidazoles, Benzoxazoles, and Benzisoxazoles." While the methodology presented does fill a gap in the existing toolbox for skeletal editing, there is a disconnect between the realities of drug discovery and the primary argument promoting this methodology for its potential application for late-stage editing during the drug development process.

Reply: We sincerely appreciate reviewer #2's time and efforts in evaluating our paper. Within the revised manuscript we added various examples outlining the potential for the structural reorganization of biologically relevant structures (see also answer to referee #1). Thus, we hope that the revised manuscript apparently shows the, in our eyes, strong connection between our strategy with the reality of drug discovery via straightforwardly generating constitutional isomers or related heteroarenes (C to N atom swap) from the same starting materials in a late-stage editing process.

The main reason of skeletal editing gradually drawing tremendous attentions in the scientific community is based on its innovative strategy to synthesize new drug candidates by enabling precise modifications of existing molecular core structures without disrupting essential functional complexity. Particularly, at late stages of the synthesis, skeletal editing minimizes the need of cost- and labor-intensive processes often associated to de novo synthesis, with thus accelerating the drug development process while keeping costs lower. For example, as described by Prof. Pennington in the review "The Necessary Nitrogen Atom: A Versatile High-Impact Design Element for Multiparameter Optimization" (Ref. 18), the replacement of a single CH group with a N atom in heteroaromatic ring systems may cause significant important effects on molecular properties and intra- and intermolecular interactions. Some literature examples showed already that the selectivity of some inhibitors was greatly improved by switching from an indole to an indazole core

via a C-to-N swap (Figure 48 in Ref. 18). Accordingly, in the revised manuscript, some drugs or drug derivatives like etodolac derivative 1ac as well as trioxsalen derivative 4q were included, which could be successfully transformed to the corresponding benzimidazole 3ac, benzisoxazole 5q and benzoxazole 6q with our strategy (see Figure 5 in the revised manuscript). In our view, these examples of the late-stage diversification of biologically relevant structures stand in good agreement to the nowadays reality in drug discovery which still heavily relates on examining large compound libraries.

Condition A involves the use of a nitrosamine reagent, which pharmaceutical companies are likely to avoid at all costs due to the heightened scrutiny from the FDA concerning nitrosamines and nitrating agents. Companies would favor a few extra steps synthesis over using such reagents, as even trace amounts pose a risk of contaminating the active pharmaceutical ingredient (API) and/or excipients.

Reply: We fully understand reviewer #2's concerns about the use of nitrosamines. However, as we illustrated above, our methodology is targeting drug discovery with synthesizing (small) amounts of novel drug analogues via late-stage C-N atom swaps of indoles or benzofurans and is primarily not intended to be used for large-scale synthesis. Larger scale synthesis might become feasible if a cheap indole containing natural product serves as the substrate. For example, the editing of tryptophan that is shown in our revised manuscript to give the unnatural indazole amino acid that might be best synthesized through the skeletal editing approach rather than through a de-novo amino acid synthesis. Anyway, our method will be likely mostly used for SAR studies where small amounts of compounds are sufficient. Thus, the strategy gives fast access to diverse architectures, but whenever a potential drug compound was discovered through such an approach, process chemists will likely always redesign de-novo syntheses for its larger scale preparation. Hence, in the SAR campaigns the amount of nitrosamine used will be always rather small. Further, in the revised manuscript we could show that N-nitrosomorpholine solutions can be formed in situ with its direct use in one-pot processes, thereby removing potential hands-on risks (see Figure C below or Figure 6a in the revised manuscript). Additionally, due to the large property differences of nitrosamines and desired products, separation of those are facile, as documented in the following chromatogram of a typical separation through MPLC (see Figure C below and revised SI page S78). Further, nitrosamines are not that stable and will likely not survive all work-up procedures, which is in contrast to transition metals that remain as impurities. Moreover, if the nitrosoamine is of serious concern for a company, we additionally present in then revised manuscript alternative ionic oxidation protocols without any nitrating agents that operate with cheap, well-established oxidants, which should satisfy FDA requirements.

Figure C One-pot process and MPLC purification.

Condition B is interesting by its simplicity; however, its practical applications are restricted due to the involvement of oxygen gas, which necessitates additional safety measures which necessitates additional safety measures as some companies require that such reactions be conducted in specialized equipment or designated laboratories which would further limit its widespread implementation.

Reply: As already mentioned above, the editing is mainly targeting SAR studies and is not primarily intended for large scale synthesis. Despite the fact that we believe that the application of oxygen is generally not very problematic for industrial companies, additional methods for the Witkop-Winterfeldt oxidation by using alternative chemical oxidants were tested for the indole and benzofuran skeletal editing. For example, cheap mCPBA performed superior in the oxidative cleavage of indoles 1j and 1k. Commercial Oxone or PCC were successfully applied for indole 1o and benzofurans 4m-4o, respectively. The detailed conditions for the respective O₂-free protocols were documented in the revised SI. Thus, we believe that we report in our revision several alternatives when respective reagents are prohibited or not compatible.

Figure 3 uses simple substrates to illustrate the scope of the reaction, which gives desired outcome with yields from low to moderate. However the scope is limited by steric effects. Most compounds of interest for pharmaceutical companies will have more than one substituent on the benzyl group, which further restricts the applicability toward drug discovery.

Reply: Indeed, the steric demand of the substituent at the indole 3-position strongly affects its reactivity as a radical acceptor. However, we found that the alternative ionic oxidation protocols circumvent this limitation by allowing the skeletal rearrangement of sterically demanding 3-substituted substrates, as we could demonstrate in the revised manuscript (see 1r-1t, 1y-1z). Indoles featuring diverse aliphatic substituents varying in steric demand from secondary cyclic rings (2-adamantyl 1j) to tertiary alkyl groups (1k) were included, which gave the indazoles 2j and 2k in 56% and 38% yield, respectively. The disubstituted benzofuran 4n bearing a phenyl group at the C2 position and an C3 isopropyl moiety engaged in a Cph to N swap in good yield. Along these examples, pimprinine, an alkaloid originally isolated from

streptomyces which has an oxazole cycle as a substituent at the C3-benzylic position afforded the corresponding indazole product **2ab** in 38% yield. Thus, diverse substrates bearing more than just one substituent at the C3-benzylic position were tolerated, further increasing the scope of our approach (see Figure D below).

Figure D Substrates bearing bulkier substituents at the C3 position and C2,C3-disubstituted congeners.

Figure 4 does offer interesting applications however focuses on simpler bioactive compounds. I would contend that the key advantage of a methodology discussed in this paper present lie with its use with more complex substrates, where restarting the sequence would be time-consuming.

Reply: We thank reviewer #2 for the valuable suggestion. In the revised manuscript we included additional more complex bioactive molecules such as dipeptides as well as tripeptides containing tryptophan residues, brevianamide F, etodolac and a psoralen derivative, which could be all converted to the corresponding indazoles and benzimidazoles (see Figure 5 in the revised manuscript). With those examples included, we believe that we could demonstrate the value of our methods to access more complex molecules where a de-novo synthesis would be more challenging or more time consuming.

It is evident considerable efforts have been invested into developing this methodology and I believe that this work may be more suitably published in the Journal of the American Chemical Society (JACS). I would recommend that the authors consider positioning this paper more as an expansion of the toolbox for skeletal editing rather than as a direct methodology for drug discovery.

Reply: We disagree with reviewer #2. Considering SAR studies starting with a lead compound, many methods published within the frame of the emerging area of skeletal editing will likely not find use. The real problem in many cases is that the structural reorganization achieved through the editing is often too large. Thus, reorganizing a lead compound will result in modified compounds that can no longer fit into the enzyme's active site. In that regard our indole to indazole and benzofuran to benzisoxazole

transformations stand out, as the CH to N swap leads to little steric reorganization, but large electronic alteration (e.g. H-bonding acceptor). Moreover, indole containing bioactive compounds are abundant and improved biological activity upon switching from indoles to indazoles has been documented in the literature (see discussion above and Ref. 18). Therefore, we believe that we are not merely presenting a synthetic tool; rather, we demonstrate that important structures can be accessed from abundant indoles. Consequently, this paper is relevant not only to synthetic chemists but also to those interested in biological applications.

Reviewer: 3

Comments

Studer and coworkers report a C to N atom swapping in indoles at C2 to arrive at indazoles and at C3 to give benzimidazoles. Furthermore, benzofurans can also be converted to benzisoxazoles or benzimidazoles. N-insertion into indoles has been demonstrated by Morandi (ref. 23, 24) and Ackermann (ref. 25) Yorimitsu has inserted boron, silicon, germanium, phosphorous and titanium into the C2–O bond of benzofurans. The authors claim that less work has been done on atom swapping in indoles and benzofurans.

The design rests on oxidative cleavage of the C2–C3 bond of indoles and benzofurans and formation of a nitrilium ion or substitution on an oxime that is formed.

The oxidative chemistry hinges on the work of Y. L. Chow (Ref. 32; note the volume number indicated in the manuscript is incorrect) and Ref. 34. Alternatively, the procedure of Witkopf-Winterfeldt oxidation starts the sequence to build ortho acyl aniline. At that point, the chemistry is relatively standard (Figure 2c) for example, following studies by Stambuli etc.

Reply: We thank reviewer #3 for his/her time and effort spent while evaluating our paper carefully. We have corrected the volume number of Ref. 32 in the revised manuscript. Both elemental steps of our strategy, the Witkop-Winterfeldt oxidation and the heterocycle formations studied by Stambuli etc. are relevant precedence but the consecutive combination for skeletal editing is underexplored. In contrast, our newly designed radical alternative procedure has, to the best of our knowledge, no precedence in literature. The cooperation of both developed strategies greatly extends the reaction scope for the conversion of indoles and delivers alternative when one approach might not be working for a substrate. For the skeletal editing of benzofurans, our novel radical pathway is highly effective over ionic oxidation routes due to the high intrinsic reactivity of the benzofuran core towards aminyl radical cations.

While the idea/concept of atom swapping is a good one, this is a multistep protocol involving workup and solvent switches. The goal of these skeletal editing reactions should be a single step or single pot procedure that will be used by others. As it stands, this is a multistep synthesis and sets the growing field of molecular editing back because it will confuse people if this is meant to represent the forefront of the field. I am not supportive of publication in the premiere journal for science such as Nature.

Reply: We respect reviewer #3's opinion. Within the field of skeletal editing most intriguing works are multistep protocols. For example, the carbon-to-nitrogen single-atom transmutation of azaarenes reported by Levin's group proceeds through oxidation of quinoline, the oxidative cleavage of quinoline N-oxide, nucleophilic attack of ammonia followed by ozonolysis to afford quinazoline (Nature 623, 77–82 (2023)). Another transformation enabling the site-specific replacement of a carbon atom of an aromatic ring with a single nitrogen atom, reported by the same group, also involves a multistep procedure including

azidation, amino-nucleophile promoted arene rearrangement, oxidant-dependent ring contraction followed by an ‘ipso’ deletion (*Science* 381, 1474–1479 (2023)). For our work instead, the main benefit lays in the selective synthesis of different constitutionally isomeric products from indole and benzofurans through one and the same intermediate (divergent synthesis). Consequently, this product divergence inevitably demands a sequential reaction strategy to allow the formation of different classes of products from a common intermediate. This diversified skeletal editing is based on the same intermediates resulting from oxidative cleavage, which greatly improved the scope of accessible products from distinctive starting materials. However, when no product divergence is demanded, conducting our manipulation as a one-pot process is also feasible, as we could showcase in our revised manuscript. Therein, as examples we display the one-pot transformation of benzofuran **1s** with in-situ formed *N*-nitrosomorpholine (see Figure E below or Figure 6a in the revised manuscript). Based on the respective “second” manipulation either the corresponding benzisoxazole **5s** (72%) or benzoxazole **6s** (55%, 3.0 mmol) were straightforwardly obtained without any workup and solvent change in between as one-pot processes.

Figure E Skeletal editing of benzofurans as one-pot processes starting from morpholine and larger scale synthesis.

Also, the work of Büchi and Tannenbaum (*JACS* 1986, 108, 4115) with follow up by Gallavardin and Franck (*RSC Adv.* 2018, 8, 13121) report very similar transformations on indoles that should, at the very least, be cited here. They also demonstrate that the concept of indole to indazole is one that has been well explored. It is used extensively already in the pharmaceutical industry even though the product has a formyl group (which is actually a useful handle).

Reply: We thank reviewer #3 for alluding to these two references. We included the mentioned works in the revised manuscript (Ref. 19 and Ref. 20). The reported transformations of indoles to indazoles acted as inspiration at the beginning of our study. However, when we tried the reported conditions applying 3-methylindole as the starting material, no product or oxime intermediate could be observed (see Figure F below). To us, this showed that the existing methodology is limited to 2,3-unsubstituted indoles. However, this stands in contrast to the fact that most of the bioactive indole derivatives and drugs carry substituents on the indole 2- and 3-positions. Together with the aspect, that we aimed for a diversified strategy towards different product through one and the same intermediate of not just indoles but also benzofurans, we considered this skeletal editing project a highly promising endeavor.

Figure F 3-Methylindole as starting material under Franck's conditions (Ref. 20).

The yields e.g., to form 2m (15%) and 3m (15%) leave a lot to be desired even though a large amount of starting materials (50%) could be recovered and the Witkopf–Winterfeldt conditions worked a little better. The ability to label with ^{15}N is a nice addition to the literature but is conceptually not new in terms of using ^{15}N labeled material which is incorporated.

Reply: *The yields of 2m and 3m are low applying the radical pathway due to the intrinsically lower reactivity of indoles compared to benzofurans as N-radical acceptors. In addition, bulkier substituents at the C3-position render the indoles less efficient radical acceptors. The recovery of a large amount of starting material shows that the transformation is not unselective and that the problem lies in the initial radical cleavage process and not in the subsequent oxime transformation. For these substrates the ionic oxidation protocols provide significantly higher yields for the initial oxidative cleavage. Further, during the revision we found a milder condition for the deformylation wherein ester bonds, like in substrate 1m (1r in the revised manuscript) are tolerated. Thus, the yield of 2r and 3r (former 2m and 3m) were greatly improved to 52% and 47%, respectively (initial protocol 15% each). For these substrates, we also decided to provide the yields obtained using the radical protocol (Method A) to document its limitations for our readers.*

Regarding the novelty of ^{15}N -labelling there is an intrinsic need to use ^{15}N -prelabeled reactants as the isotopic enrichment is generally not performed on sophisticated molecules to our knowledge. Thus, of course the strategy of using labeled precursors is not new and vastly applied in the literature, but, to the best of our knowledge, not possible differently. Additionally, our literature search showed that the ^{15}N -labeled nitrosamine used in our study was never applied in this context before.

Overall, the idea of this manuscript is a good one. Primarily because this is a multistep process, it falls short of the most impactful studies in this area of science and should be published in a more specialized journal. The supplementary material is in good shape.

Reply: *We thank reviewer #3 for the appreciation of our work and also acknowledge the positive evaluation of the Supplementary Material.*

Referee #1 (Remarks to the Author):

The authors have significantly revised the manuscript and further demonstrated the broad applicability of their method. It is particularly impressive that the method performs effectively across a diverse range of substrates, including those with valuable yet vulnerable functional groups. My previous concerns and technical suggestions have been thoroughly addressed.

Regarding the original concern raised by Reviewer 2 about the use of nitrosamines as reagents, the authors have provided additional data that effectively clarifies this issue. While I recognize that some concerns regarding safety and scalability remain, it is important to note, as the authors state in their response, that this study is focused on developing new methods to aid drug discovery rather than on revolutionizing large-scale manufacturing. To further strengthen the manuscript, I encourage the authors to include an HPLC trace of the isolated product in the Supplementary Materials after a single MPLC purification, to verify that no trace amounts of nitrosamine persist in the isolated mixture.

Reviewer 3 expressed concerns regarding the multi-step process. Although the process is indeed multi-step, I believe that medicinal chemists will appreciate the protocol for two main reasons: first, the transformation reported was previously considered unattainable, and second, the overall protocol is relatively straightforward. Moreover, while the individual steps are established in the literature, the innovative combination of these steps represents a significant advancement in expanding the previously inaccessible chemical space.

Overall, the manuscript has been considerably strengthened by the authors' revisions, and the work represents a valuable contribution to the field. I commend the authors for their thoughtful responses to the reviewers' comments.

Reply: We are grateful to the reviewer for supporting our work. The product 5s purified by MPLC and also pure N-nitrosomorpholine were both subjected to HPLC separately under the same conditions. The comparison showed that no trace of N-nitrosomorpholine was present in the isolated product. This result has been added to the revised Supplementary Information (see SI page 88).

Referee #3 (Remarks to the Author):

This is a revised manuscript from Wang, Studer and coworkers. Overall, it describes a multi-step conversion of substituted indoles and benzofurans by inserting nitrogen. The sequence is oxidative cleavage of the C2,C3 bond and then substitution onto a nitrogen or Beckmann rearrangement and then engaging a nitrilium species. As stated previously, the use of nitromorpholine in the presence of acid is a nice contribution that builds on what was known previously (Ref. 40). Overall, the authors have addressed the points raised by reviewers. However, this reviewer maintains that reporting transformations that are overall multistep by involving solvent switches is something that does not advance the field. In a journal like Nature, one expects a broad readership even among synthetic chemists. The highlighted transformation, if achieved in a single operation or through sequential additions would be used by medicinal chemists (who seem to be the target of the work). However, it is very involved as described. The recommendation is to publish in a more specialised journal. Before that, the following should be considered.

Reply: We respect Referee #3's decision. As we stated previously, there are many important works involving overall multistep transformations that greatly advanced the field of skeletal editing. As also stated by Referee #1, our protocol will be appreciated by medicinal chemists and therefore attract a broader community. It is also important to note that four different substance classes, that are all important, are available through our approach from abundant core structures (indoles and benzofurans).

a) The title should perhaps specify that this is a multistep reaction. In my opinion, the goal of skeletal editing is to accomplish single step or single pot reactions without solvent exchanges.

Reply: We have changed the title to "C-to-N atom swapping and skeletal editing in indoles and benzofurans" according to the editor's suggestion.

b) The term molecular editing is used sometimes. This nebulous term does not make sense to this reviewer.

Reply: We have replaced the term "molecular editing" to "skeletal editing" throughout in the revised manuscript.

c) C3 unsubstituted indoles and benzofurans do not work. This should be made explicitly clear. Of course, this is a complement to the Morandi work that was recently reported in the archive, but it is important to make the limitation clear. Because there are many pharmaceutical compounds etc that possess a C3 substituent, the method can still be applied to a broad range of compounds.

Reply: Referee #3 discounted that C3-unsubstituted indole 1m can be edited to the corresponding indazole 2m (see Fig. 3). 2-Methylbenzofuran without a C3 substituent was examined, but the standard condition did not provide the desired product. This result is now shown in the revised Supplementary Information (see SI page 91).

d) Figure 6: skeletal is misspelled

Reply: We thank the reviewer for alluding to this typo. We have corrected and the revised manuscript has been checked carefully.

Referee #4 (Remarks to the Author):

The reviewers' comments and the responses to the reviewers' comments have been assessed. The authors have duly complied and added the requested references for prior and contemporaneous related work. The authors have made SI additions as requested by Reviewer 1. The authors have also substantially expanded their substrate scope tables, fulfilling the requests of Reviewer 1 and Reviewer 2. In particular, the scope of the drug-like compounds in Figure 5 makes it quite believable that this chemistry can be applied to complex compounds, with yields that, as the authors argue, are certainly good enough for the medicinal chemistry space, where often only a 1-2 mg is need for a potency assay.

The authors have thoroughly addressed the concerns of Reviewer 2 about NO_x gases and O₂. I agree with the authors that NO_x and potential nitrosamines are of no concern in the discovery space, which is the audience toward which this paper is directed. In the process space, potential nitrosamine-forming species are avoided as much as possible, or else are extremely carefully controlled - however, this paper is not meant for the process chemistry space. Regarding O₂,

however, I agree with the Reviewer and disagree with the authors - med chemists are likely not going to bother setting up reactions that require O₂ gas. However, the authors do now provide alternative oxidation protocols that avoid O₂, so I consider this concern to be adequately addressed.

I understand and agree with Reviewer 2 and 3's suggestion to a journal such as JACS. There is a lack of novelty to the chemistry reactions themselves - it is more like a very good application of a sequence of chemistry. There aren't any chemistry reactions that can be called out as "the first," broadening the reader's curiosity with respect to chemical reactivity. I actually disagree with Reviewer 3's critique that the state-of-the-art in skeletal editing should be single step/pot - I have seen single-step skeletal editing approaches that are so low-yielding, and likely are in a soup of other byproduct isomers, that I doubt anyone would use them, compared to an existing, less sexy, multi-step approach that has good yields/isolations for each step. However, for a new single- or multi-step editing approach to be published in Nature, I would look for a greater degree of chemistry novelty, on top of applicability. I feel that the novelty falls short.

Reply: We are grateful to the reviewer #4 for carefully evaluating our responses to other reviewers' comments. We have to highlight that the light-mediated radical oxidative cleavage of 3-substituted indoles and 3-substituted benzofurans with N-nitrosomorpholine under mild conditions to give the corresponding ring-opened oximes has to the best of our knowledge no precedence in the literature. We therefore disagree with the statement of the referee that there "aren't any chemistry reactions that can be called out as "the first," broadening the reader's curiosity with respect to chemical reactivity".